# Optimization of an Ex-Vivo Human Skin/Vein Model for Long-Term Wound Healing Studies: Ground Preparatory Activities for the ‘Suture in Space’ Experiment Onboard the International Space Station

**DOI:** 10.3390/ijms232214123

**Published:** 2022-11-16

**Authors:** Francesca Cialdai, Stefano Bacci, Virginia Zizi, Aleandro Norfini, Michele Balsamo, Valerio Ciccone, Lucia Morbidelli, Laura Calosi, Chiara Risaliti, Lore Vanhelden, Desirée Pantalone, Daniele Bani, Monica Monici

**Affiliations:** 1ASA Research Division, ASA Campus Joint Laboratory, 50134 Florence, Italy; 2Department of Experimental and Clinical Biomedical Sciences “Mario Serio”, University of Florence, Viale Pieraccini 6, 50139 Florence, Italy; 3Imaging Platform, Department Experimental and Clinical Medicine & Joint Laboratory with Department Biology, University of Florence, Viale Pieraccini 6, 50139 Florence, Italy; 4Kayser Italia Srl, Via di Popogna 501, 57128 Livorno, Italy; 5Department Life Sciences, University of Siena, Via A. Moro 2, 53100 Siena, Italy; 6M&T Faculty, Applied Engineering and Technology, Karel de Grote University of Applied Sciences, Salesianenlaan 90, 2660 Hoboken, The Netherlands; 7Section of Surgery, Department Experimental and Clinical Medicine, University of Florence, Largo Brambilla 3, 50134 Florence, Italy

**Keywords:** wound healing, human skin, human vein, serelaxin, Zn-nonoate

## Abstract

This study is preliminary to an experiment to be performed onboard the International Space Station (ISS) and on Earth to investigate how low gravity influences the healing of sutured human skin and vein wounds. Its objective was to ascertain whether these tissue explants could be maintained to be viable ex vivo for long periods of time, mimicking the experimental conditions onboard the ISS. We developed an automated tissue culture chamber, reproducing and monitoring the physiological tensile forces over time, and a culture medium enriched with serelaxin (60 ng/mL) and (Zn(PipNONO)Cl) (28 ng/mL), known to extend viability of explanted organs for transplantation. The results show that the human skin and vein specimens remained viable for more than 4 weeks, with no substantial signs of damage in their tissues and cells. As a further clue about cell viability, some typical events associated with wound repair were observed in the tissue areas close to the wound, namely remodeling of collagen fibers in the papillary dermis and of elastic fibers in the vein wall, proliferation of keratinocyte stem cells, and expression of the endothelial functional markers eNOS and FGF-2. These findings validate the suitability of this new ex vivo organ culture system for wound healing studies, not only for the scheduled space experiment but also for applications on Earth, such as drug discovery purposes.

## 1. Introduction

Long-term space missions to bring astronauts beyond the Earth’s orbit, to explore the closer planets, have become a feasible objective for the near future. This has prompted a surge in biomedical research in order to identify the possible risks and health issues of exposing the human body to prolonged micro/low gravity conditions and to set up appropriate countermeasures, especially taking into account the limited availability of medical resources onboard a spaceship. Studies on astronauts returning from missions onboard the International Space Station (ISS) have shown that prolonged exposure to microgravity can impair tissue homeostasis, thus negatively influencing bone and skeletal muscle, hematopoiesis and immune response. Although studies on wound healing in humans or in human tissues have never been conducted in space until now, the pathophysiological alterations caused by space flight could impair the body’s resilience to injuries [1,2,3]. In particular, the susceptibility of astronauts to trauma due to particular working needs and conditions makes the impaired tissue response to wounds a major reason for concern [4], as well as a priority subject for biomedical research funded by the national and international space agencies. In this context, our research group is involved in an international, multidisciplinary research project to investigate how human tissues, particularly the skin and blood vessels, can adapt to microgravity conditions and how wound repair may be influenced [5]. This study was selected by the European Space Agency (ESA) to be carried out at the ISS and in parallel on Earth, tentatively in 2022 (ESA-AO-ILSRA-2014). It exploits two ex vivo human organ models, namely whole-thickness skin and saphenous vein bearing a standardized sutured wound, prepared from samples donated by volunteer patients subjected to mammary plastic or vascular bypass surgery. To ensure tissue viability throughout the experiment (4 weeks), we have developed an automated tissue culture chamber–which fits the ESA Biolab facility inside the Columbus module onboard the ISS–and a new tissue culture technique that combines biochemical and biophysical factors. The culture chamber is equipped with a device that can model physiological tensile strength in the tissues and monitor its changes throughout the experiment, thus enabling the study of tissue and suture mechanical properties. Mechanical factors are involved in the regulation of many biological processes, wound healing included, which are crucial for maintaining tissue homeostasis. Therefore, the modeling of physiological and mechanical factors improves tissue culture survival. Moreover, new long-term culture media have been created. They are based on standardized media, supplemented with substances previously used to extend the viability of explanted organs scheduled for transplantation, namely serelaxin and (Zn(PipNONO)Cl) [5]. Serelaxin, the recombinant form of human H2 relaxin hormone suitable for pharmaceutical use, has been shown to induce protection against ischemia–reperfusion injury by reducing cellular oxidative damage, apoptosis and inflammation [6,7]. By the use of similar mechanisms, serelaxin has been shown to extend the lifespan of liver and lungs to be transplanted [8]. Zn-metallononoate (Zn(PipNONO)Cl), a nitric oxide (NO)-releasing molecule (Noxamet Ltd., Milan, Italy), has been shown to promote endothelial cell survival and to induce the activation of H2S-dependent signaling pathways, resulting in potent antioxidant and tissue-trophic effects in in vitro culture [9,10]. As a mandatory preliminary step to the true experiment, we have performed the present study to ascertain whether our model is adequate to the purpose, and particularly whether the wound healing process can take place, in full or in part, even in these long-term skin and vein explants. To achieve this aim, at the end of the experimental period, key indicators of cell function and tissue remodeling were investigated in tissue regions taken either in close proximity to or at a distance from the surgical wound.

## 2. Results

### 2.1. EU Set-Upg

The Experimental Units (EU) used for the experiments are composed of a stainless steel frame, allowing the explanted skin specimens to be stitched onto micro-adjustable support brackets made of biocompatible plastic, a culture chamber filled with enriched incubation medium and a reservoir containing fresh medium to replace the exhausted medium in the culture chamber via a peristaltic pump (Figure 1), designed to prevent formation of air bubbles within the chamber. The body of the culture chamber and reservoir are made of sterilizable, biocompatible plastic. External air can readily exchange with the tissue samples via a gas-permeable silicone membrane, which also prevents bacterial contamination. In previous experiments, we observed that oxygen consumption varied depending on the tissue type, preservation conditions, time since collection, temperature and salinity. In general, oxygen consumption in tissue samples was ≤0.5 µmol/mL/h (with salinity ranging from 0 to 10 g/L). The thickness (0.2 mm) and exchange surface (3350 mm^2^) of the membranes used in the EU has an oxygen permeability of 16.2 cm^3^/h, which largely fulfills the metabolic needs of the incubated tissues [5]. The mounted skin or vein specimen is connected to a load cell (Burster Srl., Bergamo, Italy) capable of continuously measuring and recording the tensile strength in a 0–20 N range during the whole experiment by means of dedicated software developed by Kayser Italia. Electronics, including a microcontroller, are assembled within the EU in an Advanced Experiment Container (AEC), which has been sized and designed to fit and plug the Biolab slots onboard the ISS (OHB, Bremen, Germany). In this way, the EU can be interfaced with the Biolab controllers and pre-programmed software to receive commands, e.g., pump start/stop, load cell operation, data acquisition, etc. For the present experiment, the AECs were connected to a Biolab interface simulator. Both the experimental hardware and software worked properly for the entire experimental period (a minimum of 28 days), performing turnover of the incubation medium and recordings of tensile strength at the set time points with no need for external adjustments.

### 2.2. Histology

Light microscopic examination of hematoxylin and eosin (H&E)-stained histological sections of the skin specimens after long-term incubation in the enriched incubation medium and a freeze-and-thaw cycle showed a substantially preserved architecture of the epidermis, as well as the papillary and reticular dermis (Figure 2). In all of the specimens the surgical wound was still clearly detectable, although the epidermal layer at the wound edges appeared thicker and with more pronounced ridges than it did at a distance from the wound (Figure 2A). At higher magnifications, epidermal keratinocytes showed a normal appearance, the only detectable abnormality being a slight cytoplasmic vacuolation in some supra-basal cells; in some specimens, melanin pigmentation was still visible in the basal layer. Blood microvessels and dermal stromal cells also appeared well preserved (Figure 2B). By comparison, a skin specimen incubated in a standard culture medium not enriched with serelaxin and (Zn(PipNONO)Cl) showed prominent abnormalities, namely diffuse cytoplasmic swelling of keratinocytes and detachment of the epidermis from the underlying basement membrane (Figure 2C). Dermal blood vessels were barely detectable and stromal cells often showed hyperchromatic, picnotic nuclei.

Histological examination of H&E-stained saphenous vein sections subjected to the same incubation protocol also showed a substantially preserved architecture of the vascular wall layers or tunicae, the intima, media and adventitia (Figure 3). In particular, at higher magnification, a nearly continuous endothelium was observed, together with normally appearing smooth muscle and stromal cells (Figure 3B). By comparison, a vein specimen incubated in a standard non-enriched culture medium showed a nearly complete disappearance of the endothelial layer (Figure 3C).

### 2.3. Morphometric Analysis of Connective Tissue Fibers

Histological sections of the same skin specimens were then analyzed by histochemistry and morphometry to investigate possible differences in dermal collagen and elastic fibers that could be related to the wound healing process. The percent surface area of the meshwork of picrosirius red (PSR)-stained collagen fibers was significantly decreased in the papillary dermis close to the surgical wound, compared to that at a distance from it, suggesting that collagen remodeling had occurred. On the other hand, no significant differences were found in the percent surface area of the thicker collagen fibers in the reticular dermis, regardless their proximity to the wound (Figure 4). Similarly, in the reticular dermis, no significant regional differences of paraldehyde–-fuchsin (PAF)-stained elastic fibers were detected (Figure 5). In the papillary dermis, elastic fibers were too few for a reliable morphometric analysis.

Histochemistry and morphometry of the vein sections showed no significant differences in the percent surface area of PSR-stained collagen fibers and PAF-stained elastic fibers between the tunica adventitia measured in tissue areas close to or far from the surgical wound (Figure 6 and Figure 7). Instead, a slight but significant reduction of the percent surface area of the elastic fibers in the tunica media was detected in proximity to the wound (Figure 7). This finding is consistent with the occurrence of extracellular matrix remodeling.

### 2.4. Blood Microvessels and Stromal Cells

Small-sized blood vessels, already observed in the H&E-stained sections, were specifically identified by fluorescein isothiocyanate (FITC)-labeled *Ulex europaeus* agglutinin (UEA) lectin and anti-α-smooth muscle actin (α-SMA) antibodies (Figure 8A). They were mostly located in the papillary and upper reticular dermis and did not show any visual differences between the tissue areas close to and far from the wound. Mast cells, identified by FITC-labeled avidin, were found in the upper dermis, especially along blood vessels and around adnexa (Figure 8B,C), as well as in the tunica adventitia of the vein samples. Visually, those located in proximity to the wound appeared to contain fewer fluorescent granules, suggesting that cell activation and granule release had occurred. Both of the dermal layers harbored several spindle-shaped or stellate fibroblasts expressing the activation marker heat-shock protein 47 (HSP47), which was also expressed by epidermal keratinocytes. It was also noted that activated fibroblasts appeared to be more numerous in the dermal areas close to the wound (Figure 8D,E). On the other hand, HSP47-positive fibroblasts were only seldom encountered in the vein samples. No α-SMA-positive stromal cells identifiable as myofibroblasts were detected in any dermal or vein wall areas.

### 2.5. Proliferating Epidermal Keratinocytes

Histological sections of the same skin specimens were then immunolabeled with anti-Ki67 antibodies to identify proliferating cells. The percentage of Ki67-positive keratinocytes over total cells was slightly, albeit significantly, higher in the epidermis of areas close to the surgical wound than at a distance from it (Figure 9). As expected, Ki67-positive cells were mainly located in the basal layer of the epidermis and at the periphery of the hair follicle sheath, likely at the level of the so-called ‘bulge’, which is known to harbor epidermal stem cells [11]. Scattered Ki67-positive cells were seldom found in the papillary dermis.

### 2.6. Markers of Tissue Integrity and Functions

Protein lysates from skin and vein tissues taken far from or close to the surgical wound were analyzed by western blotting to evaluate the expression of two key markers of endothelial integrity and function, fibroblast growth factor 2 (FGF-2) and endothelial nitric oxide synthase (eNOS), as well as the inducible inflammatory and tissue remodeling marker inducible nitric oxide synthase (iNOS) (Figure 10). These molecules were detectable and measurable (as ADU) in all the samples, thus allowing reliable comparisons. In the skin samples, the levels of eNOS and FGF-2 were significantly higher in the specimens taken in proximity to than distant from to the wound, while iNOS was unchanged (Figure 10A,B). In the vein samples, eNOS and FGF-2 also attained higher levels in the specimens taken in proximity to than distant to the wound, the eNOS differences reaching statistical significance, whereas no substantial changes were detected for iNOS (Figure 10C,D).

## 3. Discussion

The present study offers experimental evidence that, under the described culture conditions, ex vivo human skin and vein specimens can be maintained in a viable state for a long time, enabling them to activate some tissue-specific steps of wound healing. In particular, the findings on the epidermis and dermis from areas located in close proximity to the sutured surgical wound have shown activation of the keratinocyte stem cell compartment and of dermal fibroblasts, as judged by Ki67 and HSP47 immunofluorescent detection, accompanied by remodeling of the superficial collagen fiber network. Both of these features are typical of the wound healing process, as they predispose to epidermal cell migration from the wound margins to achieve wound closure [12,13,14]. In this context, the observed expression of HSP47 by both epidermal cells and dermal fibroblasts is particularly relevant: this collagen-binding chaperonin has been reported to be induced by stress conditions, when it plays a major role in the control of collagen biosynthesis by preventing the secretion of abnormal procollagen [15]. Due to its collagen regulatory properties, HSP47 has been shown to be up-regulated at both the transcriptional and translational levels during skin wound healing [16]. Its detection in the tissues of the studied skin specimens is a major piece of evidence in favor of the suitability of the present ex vivo model for wound healing studies. Finally, the fact that mast cells show signs of degranulation, accounting for cell activation, in the vicinity of the wound suggests that the activity of fibroblasts can also be influenced by their mediators [17].

The data collected from the vein model also suggest that some of the known events of vascular wall remodeling during wound healing have taken place: in particular, the reduction of the elastic fiber network in the tunica media is consistent with extracellular matrix degradation, which precedes its replacement with collagen (scarring) [18]. The slight increase in the endothelial functional markers FGF-2 and eNOS and in the smooth muscle activation marker iNOS in the specimens close to the wound also point to this conclusion.

Key features of our model that can account for its successful outcome are restoration of physiological skin and vein tensile strength and enrichment of the incubation medium with tissue protective substances. Concerning the first point, it has been demonstrated that survival of explanted tissues was impaired if free biopsies were allowed to shrink, while it was improved if biopsies were fixed onto a support to maintain the original tissue geometry [19]. Our EU allows the stitching of excised specimens onto an adjustable frame capable of restoring approximately their initial size and physiological intrinsic mechanical forces, as well as the measurement and recording of their changes during the surgical wounding, suturing and healing process. Concerning the second point, a series of studies on cellular, isolated organ and animal models of ischemia/hypoxia followed by reperfusion/reoxygenation have demonstrated that serelaxin acts at multiple levels in the complex network of the mechanisms of oxidative damage during post-ischemic reoxygenation, exerting marked protective effects [6,7]. With this background, since oxidative stress is also a key factor underlying the progressive damage and viability loss of explanted organs, serelaxin has been exploited as a protective supplement to the maintenance medium of isolated organs before transplantation, with positive results [8]. Moreover, oxidative stress has been identified as a pathogenic factor of microgravity-induced tissue damage [20]; thus, serelaxin could be beneficial in more ways for tissue maintenance in the planned experiments onboard the ISS. For the same reasons, metal–nonoates, which behave as potent antioxidants by a molecular mechanism involving NO release and H2S increase [9,10], can be helpful for this purpose. Indeed, the present findings confirm that serelaxin and (Zn(PipNONO)Cl]) are useful pharmacological tools to extend the viability of the isolated skin and blood vessel explants during the whole experimental period, as required for the planned studies onboard the ISS. More generally, by preventing oxidative damage, this culture medium, supplemented with serelaxin and and (Zn(PipNONO)Cl), could be particularly suitable for culturing biological tissue and 3D tissue constructs onboard space vehicles/stations in future deep-space missions.

A limitation of our model directly relates to the prolonged ex vivo conditions. In particular, although the EU allowed an adequate O_2_ and metabolite supply for the metabolic needs of the skin and vein specimens [5], the absence of blood perfusion deprived the tissues of the substantial contribution of blood-borne inflammatory cells and mesenchymal stem cells. Together with resident cells, these play an important role in wound healing by the secretion of cytokines and growth factors that mediate inflammation and fibroblast–myofibroblast trans-differentiation [21,22,23]. The absence of α-SMA-positive myofibroblasts in proximity to the surgical wound can be explained by the lack of cytokine-mediated (e.g., TGF-β) induction by inflammatory cells which, in normal in vivo conditions, infiltrate the wounded tissue [24,25].

Another cause for the ostensible absence of myofibroblasts could be the lack of mechanical stresses that are normally present and continuously operating in vivo, for example, due to joint movements. It is well known that myofibroblast differentiation is strongly stimulated by mechanical forces [26] and fibrotic scars generally occur in body regions particularly exposed to mechanical stress [27]. Our future studies will focus on the application of discontinuous mechanical stresses to our ex vivo models, which also appear suitable for studying how mechanical forces may influence wound healing.

Because of the above-described limitation, as well as the fact that the remodeling phase of wound healing takes place beyond the time frame of the current ex vivo model, only part of the complex cellular and molecular mechanisms of wound healing can be investigated. However, the described ex vivo long-term human skin and vein models represent useful experimental tools to dissect and better understand the mechanisms of wound healing. For instance, the modest or absent inflammatory reaction may be an opportunity to study tissue regeneration. In fact, it has been well assessed that scarring is controlled by myofibroblasts differentiated and activated by a strong inflammatory reaction to tissue injury, while regeneration is instead associated with poor inflammation [28].

In conclusion, beside the specific goal of carrying out the planned experiment onboard the ISS, this whole-tissue human skin/vein model may represent a valuable alternative to in vitro keratinocyte/fibroblast co-cultures or in vivo rodent models for the study of wound healing and, possibly, a valuable option for drug development purposes.

## 4. Materials and Methods

### 4.1. Specimen Sampling and Handling

Skin biopsies (2 cm length) taken at surgery (*n* = 4) were carefully purged of subcutaneous fat, rinsed in PBS and stitched with 3.0 non-absorbable nylon suture onto square frames specifically developed to stretch the tissue, simulating physiological tensile strength, and to monitor its changes during wound healing (Kayser Italia Ltd., Livorno, Italy). Then, the frames with the skin biopsies were placed in transport containers filled with modified RPMI and kept at 4 °C for 17 days to simulate the time period between collection of biopsies at Careggi University Hospital (Florence, Italy) and preparation of sutured wound models at the launch site (Kennedy Space Center, Cape Canaveral, FL, USA), including the sample shipment time. A similar procedure was used to prepare saphenous vein biopsies, about 2 cm long and with a 0.5 cm diameter, which were stitched with 6.0 non-absorbable polypropylene suture onto the ends of the frames and cultured in the conditions described below. At day 18, to simulate the preparation of the wound models at the launch site, 1 cm long incisions were made on the skin biopsies with a scalpel and these were then sutured with 3.0 non-absorbable nylon thread to reproduce a surgically closed skin wound. The vein samples were cut transversely in the middle and sutured with 6.0 non-absorbable polypropylene thread to reproduce an end-to-end vascular anastomosis. Then, both of the sutured wound models were put into the in-flight experiment hardware: each sample was placed in the culture chamber of an experiment unit (EU), filled with modified DMEM incubation medium. The EUs were placed in pairs (1 skin and 1 vessel) in experimental containers (ECs) and kept at 24 °C for 6 h (to simulate the handover from the launch site to the ISS). Finally, the ECs were incubated at 32 °C for 9 to 12 days (to simulate the in-flight experiment on board the ISS). The actual in-flight experiment will last from 4 to 9 days (the various samples will be retrieved at different times to evaluate different phases of wound healing). In the present ground simulations, the longest incubation time was chosen to ensure adequate viability of the experimental models. At the end of experiment, the specimens were frozen at −80 °C directly in the medium-filled culture chambers. After 2–10 weeks (to simulate the return to Earth and trans-continental transport to the PI’s laboratory) each frozen sample was cut into 2 halves, orthogonally to the surgical wound: one was gently thawed by immersion in Immunofix (Bio-Optica, Milan, Italy) formaldehyde-based fixative solution at 4 °C and then processed for conventional light microscopy, i.e., dehydrated, embedded in paraffin and cut into 5 μm-thick sections, while the other was kept frozen for molecular analyses.

### 4.2. Bioreactor Development

The bioreactor has been developed by Kayser Italia, following the requirements indicated by the scientific team. The culture chamber, the frame conferring mechanical support to the tissue specimens, the system including the medium reservoir and the electronically controlled peristaltic pump for medium circulation, the gas-permeable membrane, the electronically controlled load cell connected to the frame for modeling the physiological tensile strength in the tissue, and the monitoring of this throughout the culture period have been developed as previously described [5]. All of the bioreactor components in contact with the tissues and culture media are biocompatible.

### 4.3. Enriched Long-Term Incubation Media

Two modified incubation media were used in the experiments, namely RPMI (Sigma-Aldrich, Milan, Italy) for maintenance of the sample at 4 °C and DMEM (Sigma-Aldrich) for incubation of the samples at 24–32 °C. RPMI was supplemented with 120 μg/mL lincomycin (Pfizer, Latina, Italy), 10 μg/mL colistin (Accord Healthcare Italia S.r.l., Milan, Italy) and 50 μg/mL vancomycin (Hikma Italia S.p.a., Pavia, Italy); DMEM was supplemented with 8 μL/mL 20% bovine serum albumin, 0.4 μg/mL hydrocortisone (Sanofi S.r.l., Anagni, Italy), 0.12 UI/mL insulin (Ely Lilly, Sesto Fiorentino, Italy), 100 UI/mL G penicillin (Sigma-Aldrich), 20 μg/mL gentamycin (L.F.M., Milan, Italy), 1 μg/mL amphotericin B and 50 μg/mL ascorbic acid (Sigma-Aldrich). Both media were enriched with substances previously used to protect explanted organs for transplantation purposes, namely serelaxin (60 ng/mL) and (Zn(PipNONO)Cl) (28 ng/mL), with the aim of extending the viability of the ex vivo organ specimens during the experimental period. The currently used concentrations of serelaxin and (Zn(PipNONO)Cl) have been chosen based on those that exerted significant tissue protective effects in the above cited studies [8,9,10].

### 4.4. Histology and Morphometric Evaluation of Collagen and Elastic Fibers

Histological sections, 5 μm thick, were cut from the paraffin-embedded samples. Some of them were stained with H&E and observed using light microscopy. Histological images were acquired using a microscope equipped with a Visicam TC10 tablet camera (WWR International, Milan, Italy). Others were stained with 0.2% PSR for 60 min., a histochemical method specific for collagen fibers. Staining of the sections was performed in a single session, to minimize artifactual differences. In each skin specimen, 2 photomicrographs, including the papillary and reticular dermis, were randomly taken from areas in close proximity to or distant from (3–5 mm) the surgical wound; in each vein specimen, 2 photomicrographs of the tunica media and 2 of the tunica adventitia were randomly taken from areas in close proximity to and distant from (3–5 mm) the surgical wound. In both instances, a Nikon DS F12 CCD camera connected to a Nikon Eclipse E200 light microscope with a 40× objective (each micrograph: 57,700 μm^2^) was used. On each image, 4 regions of interest (ROI), 1500 μm^2^ each, were randomly chosen: here, the surface area of PSR-stained collagen fibers was selected by thresholding (to exclude the PSR-negative amorphous ground substance and cells) and measured using ImageJ 1.53 k software (http://imagej.nih.gov/ij (accessed on 2 October 2021). Values are expressed as percent area of collagen fibers over total tissue area. In the vein samples, only the tunica adventitia close to or far from the wound was selected for the measurements, since collagen fibers are chiefly present in this layer. Another series of histological sections was used to assess the percent area of elastic fibers, stained histochemically with 0.5% PAF for 5 min, applying a similar morphometrical method and sampling procedure. In the skin samples, only the reticular dermis close to or far from the wound was selected for the measurements, since elastic fibers in the papillary dermis were too few to give reliable results.

### 4.5. Fluorescent/Immunofluorescent Detection of Mast Cells, Blood Vessels and Fibroblasts

Histological sections (5 μm thick) were cut from te paraffin-embedded skin and vein samples and used for detection of different stromal cell types using specific fluorescent markers, as follows: FITC-labelled avidin (1:400) for mast cells [29]; FITC-labelled UEA lectin (1:10) for blood vessels [30]; rabbit polyclonal anti-HSP47 (1:50 Abcam, Milan, Italy) followed by FITC-labelled goat anti-rabbit antibodies (1:32 Abcam) for activated fibroblasts [31]; and goat polyclonal anti-α-SMA (1:400 Abcam) followed by FITC-labelled rabbit anti-goat antibodies (1:175 Abcam) for both blood vessels and myofibroblasts [32]. In some slides, nuclei were counterstained in red with propidium iodide. Before each immunolabeling, antigen retrieval was performed using 0.1 M citrate buffer at 96 °C for 10 min. The fluorescent markers and the primary antisera were applied overnight at 4 °C, and the secondary antisera for 2 h at 37 °C. Omission of the primary antibody was used as a negative control for the immunofluorescence reactions. The labelled sections were viewed and photographed using a Zeiss Axioskop UV microscope equipped with a digital camera and Axiovision 4 software (Zeiss, Jena, Germany) or a Leica TCS SP5 confocal microscope. Unless otherwise stated, all reagents were from Sigma-Aldrich.

### 4.6. Evaluation of Proliferating Epidermal Keratinocytes

Migration of newly formed keratinocytes to fill the skin defect is a key early step of wound healing [12,14]. To assess whether this phenomenon also occurred in our ex vivo model, a series of sections from the paraffin-embedded specimens was immunostained to reveal the Ki67 nuclear proliferation antigen. Briefly, the sections were subjected to antigen retrieval as described above, incubated overnight at 4 °C in rabbit polyclonal anti-Ki-67 antiserum (Sigma-Aldrich), and diluted at a ratio of 1:50 in PBS with 3% bovine serum albumin. An immune reaction was revealed by sequential incubation (at room temperature) in biotinylated goat anti-rabbit antiserum (Thermo Fisher Scientific, Milan, Italy; 1:600, 30 min), avidin/peroxidase complex (Thermo Fisher Scientific, 10 min), and DAB substrate kit (Abcam, Cambridge, UK; 5 min) as chromogen. Nuclei were counterstained with hematoxylin. In each skin specimen, the percentage of Ki67-positive nuclei over total nuclei of basal/suprabasal keratinocytes was counted by a trained observer directly from a light microscope with a ×40 objective, in at least 2 microscopic fields, in close proximity to or distant from the surgical wound.

### 4.7. Western Blotting

Western blotting analysis was performed on frozen samples of skin and vein, as described [33], taken in proximity to (close) or distant from (far, 3–5 mm) the surgical wound. Protein extraction was achieved upon disruption and homogenization of the specimens using a TissuesLyser II (Qiagen, Germantown, MD, USA). Samples were frozen/unfrozen twice in liquid nitrogen and then sonicated on ice for a total of 2 min, with a 15 s run and 15 s pause to limit heating. Tissue lysates were centrifuged at 16,000× *g* for 20 min at 4 °C and the supernatants were then collected. Protein concentration was determined using the Bradford method. Electrophoresis (50 μg of protein/sample) was carried out in 4–12% Bis-Tris Gels (Life Technologies, Carlsbad, CA, USA). Proteins were then blotted onto nitrocellulose membranes and incubated overnight with the following primary antibodies: anti-eNOS (mouse, 1:1000, cat. no. 612656, BD Transduction Laboratories, Franklin Lakes, NJ, USA), anti-iNOS (rabbit, 1: 1:500, cat. no. sc-651, Santa Cruz Biotechnology, Dallas, TX, USA), anti-FGF-2 (mouse, 1:500, cat. no. 05-118, Merck KGaA, Darmstadt, Germany), and anti-β-actin (mouse, 1:10,000, cat. no. MABT825, Merck KGaA). Immune reactions were detected by an enhanced chemiluminescence system (Biorad, Hercules, CA, USA). The results were normalized to those obtained with anti-β-actin antibodies (mouse, 1:1000, cat. no. 612656, BD Transduction Laboratories). Immunoblots were analyzed by densitometry using Fiji software (64-bit Java 1.8.0_172), and the results, expressed as arbitrary density units (ADU), were normalized to β-actin.

### 4.8. Statistical Analysis

The experimental values were expressed as the mean ± s.e.m. of the 4 different skin or vein specimens, each assumed as the test unit. Statistical comparison of data measured close to or far from the wound was performed by using Student’s t test for unpaired values, assuming *p* ≤ 0.05 as significant. Calculations and graphical rendering was carried out with Prism 5.0 software (GraphPad Dotmatics, Boston, MA, USA).

## Figures and Tables

**Figure 1 ijms-23-14123-f001:**
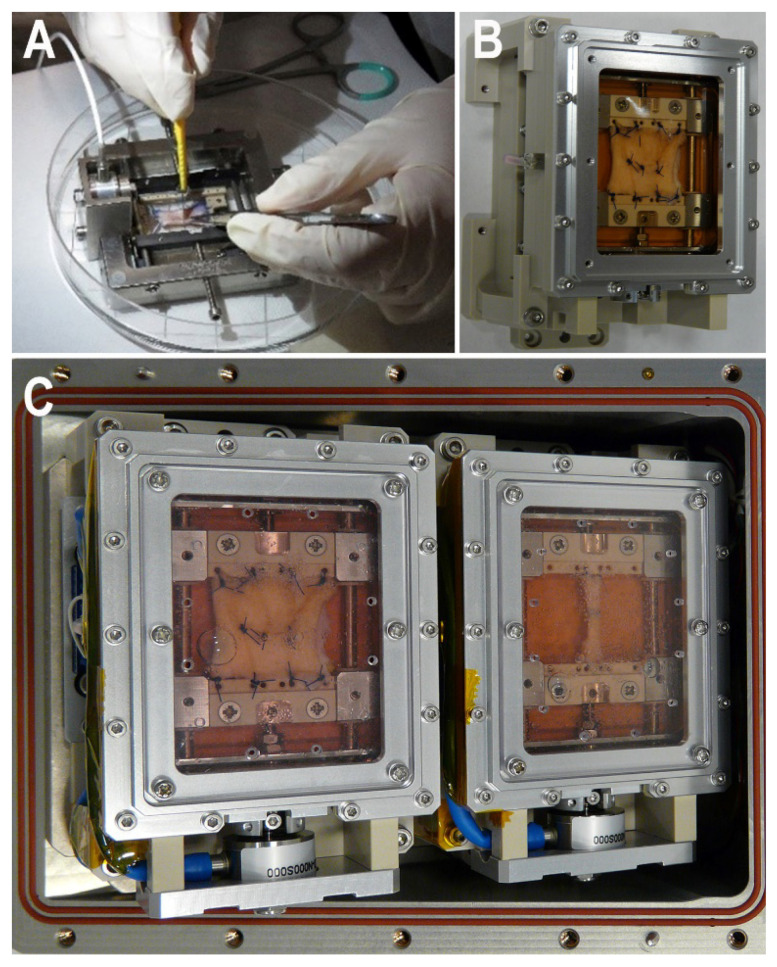
Representative photographs of the Experimental Unit (EU). (**A**) Detail of the stainless steel frame during surgical manipulation; (**B**) EU with a mounted skin specimen and a sutured surgical wound; (**C**) Two operating EUs assembled in an Advanced Experiment Container to be inserted into the Biolab facility of the Columbus module at the ISS.

**Figure 2 ijms-23-14123-f002:**
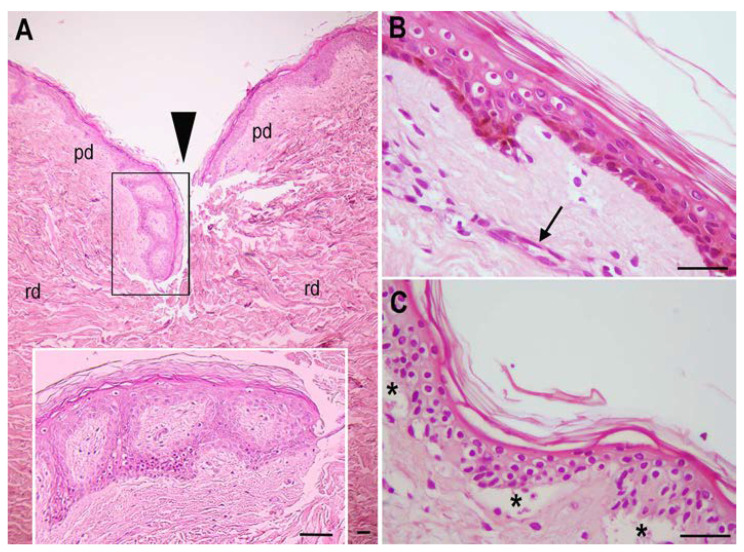
Representative histological features of the skin specimens after long-term incubation, freezing and thawing. (**A**) Transverse section across the surgical wound shows the epidermis nearby with pronounced rete ridges (inset); pd: papillary dermis, rd: reticular dermis. (**B**) High magnification of a skin specimen incubated in enriched medium showing a substantially normal epidermis, with brown melanin pigment in the basal layer and scattered keratinocytes with vacuolated cytoplasms, and preserved blood capillaries (arrow). (**C**) Same magnification of a skin specimen incubated in non-enriched medium showing diffuse keratinocyte vacuolation and detachment of the epidermis from the basement membrane (asterisks). H&E staining; bars = 100 μm.

**Figure 3 ijms-23-14123-f003:**
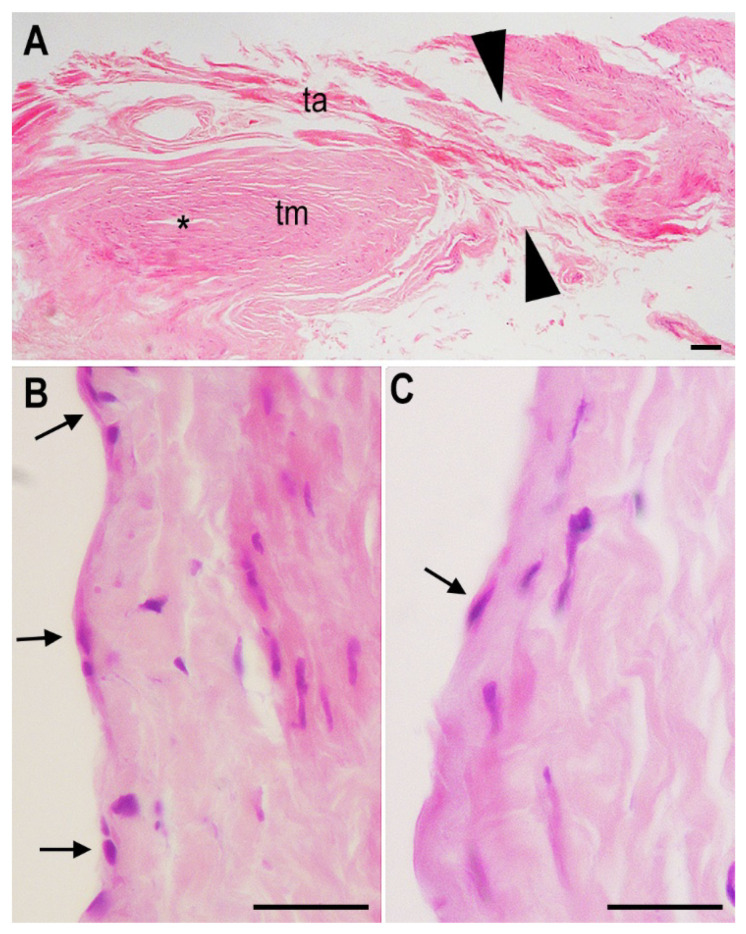
Representative histological features of the saphenous vein specimens after long-term incubation, freezing and thawing. (**A**) Longitudinal section across the surgical wound, indicated by arrowheads; asterisk: lumen, tm: tunica media, ta: tunica adventitia. (**B**) High magnification of a vein specimen incubated in enriched medium showing a nearly continuous layer of endothelial cells (arrows). (**C**) Same magnification of a vein specimen incubated in non-enriched medium showing diffuse endothelial loss, with only scattered residual endothelial cells (arrow). H&E staining; bars = 100 μm.

**Figure 4 ijms-23-14123-f004:**
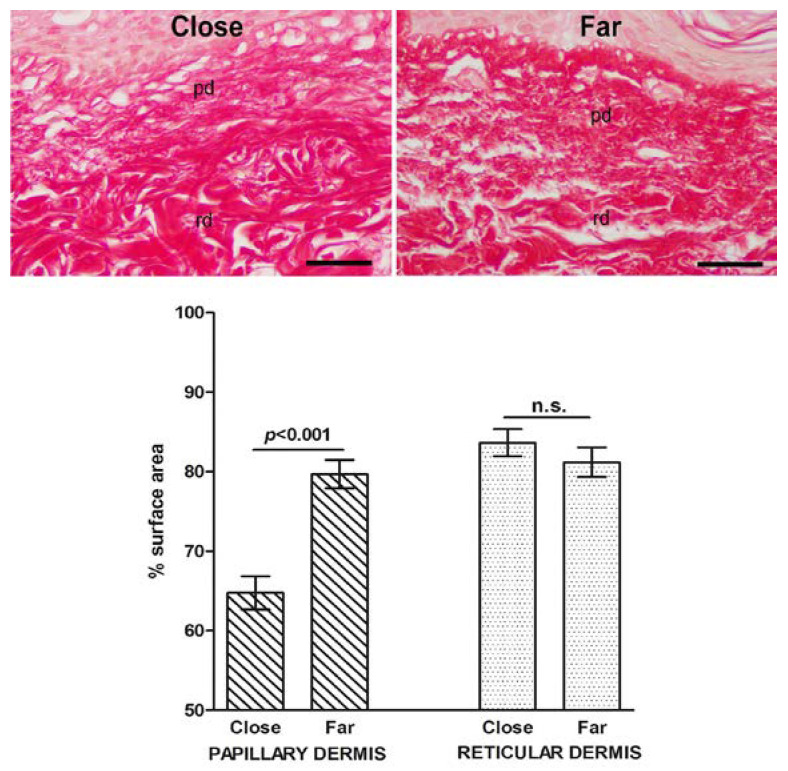
Light microscopic appearance and morphometry of collagen fibers from skin areas located close to and far from (3–5 mm) the surgical wound; pd: papillary dermis, rd: reticular dermis. The collagen fiber meshwork is slightly, albeit significantly, reduced in proximity to the wound. PSR staining; bars = 100 μm; values are mean ± s.e.m., *n* = 4, n.s. not significant.

**Figure 5 ijms-23-14123-f005:**
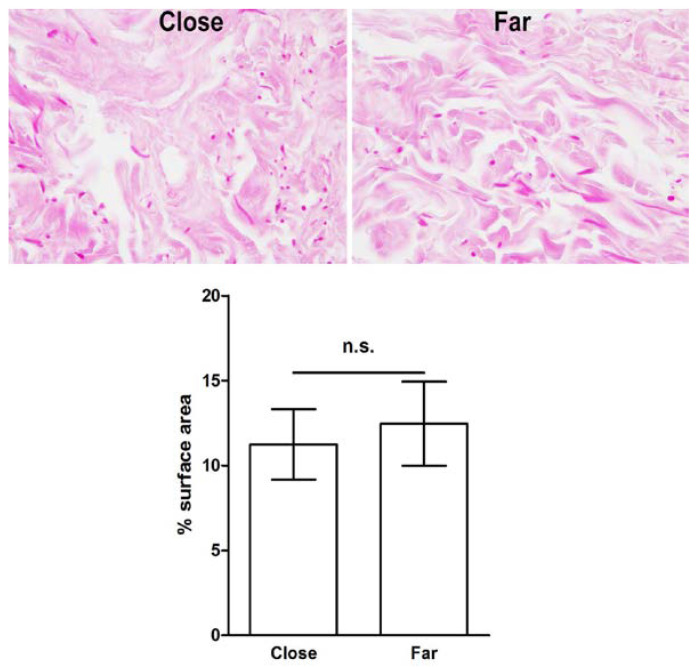
Light microscopic appearance and morphometry of elastic fibers in the reticular dermis from skin areas located close to and far from (3–5 mm) the surgical wound. No substantial differences can be seen or measured. PAF staining; bars = 100 μm; values are mean ± s.e.m., *n* = 4, n.s. not significant.

**Figure 6 ijms-23-14123-f006:**
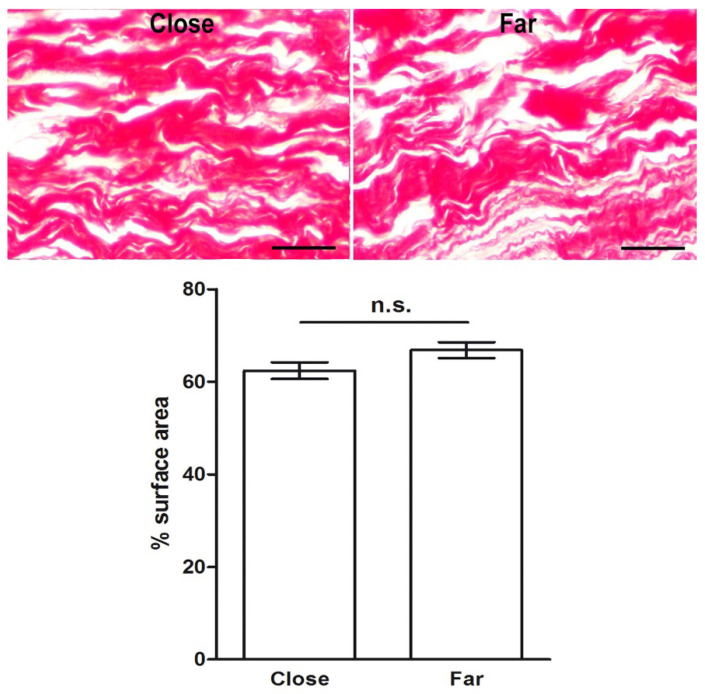
Light microscopic appearance and morphometry of collagen fibers from vein areas of the tunica adventitia located close to and far away (3–5 mm) the surgical wound. No substantial differences can be seen or measured. PSR staining; bars = 100 μm; values are mean ± s.e.m., *n* = 4, n.s. not significant.

**Figure 7 ijms-23-14123-f007:**
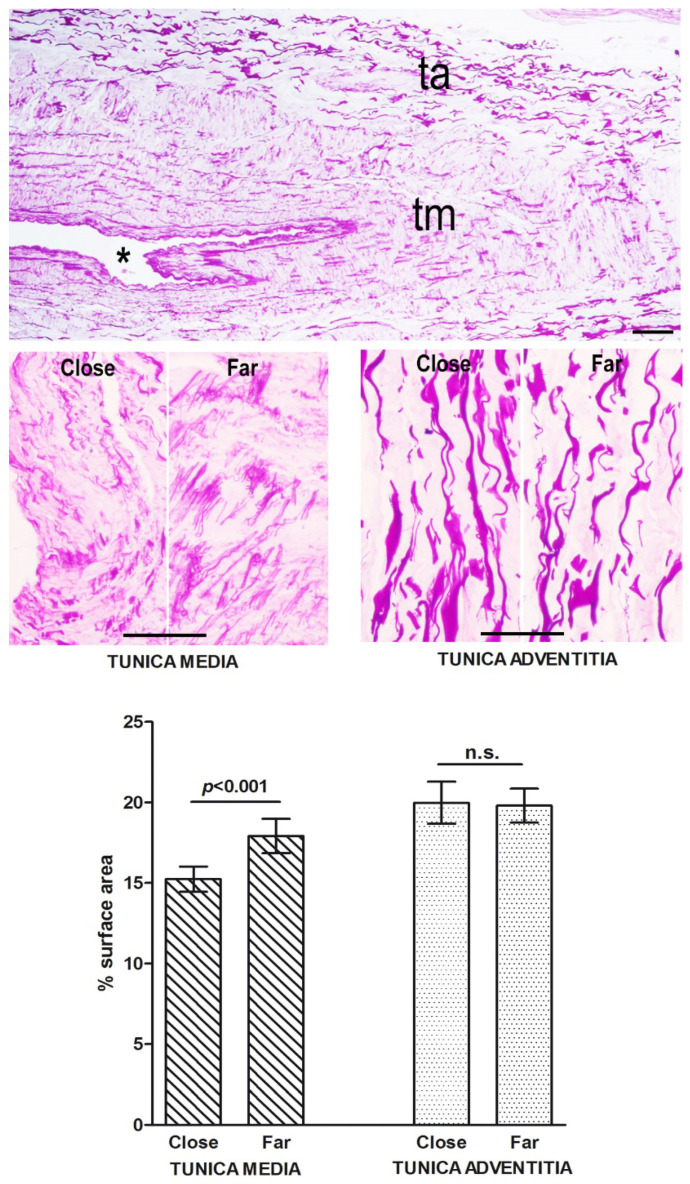
Light microscopic appearance and morphometry of elastic fibers from vein wall areas located close to and far from (3–5 mm) the surgical wound. No substantial differences can be seen or measured in the tunica adventitia (ta), whereas in the tunica media (tm) the elastic fiber meshwork is slightly, albeit significantly, reduced in proximity to the wound. * lumen. PAF staining; bars = 100 μm; values are mean ± s.e.m., *n* = 4, n.s. not significant.

**Figure 8 ijms-23-14123-f008:**
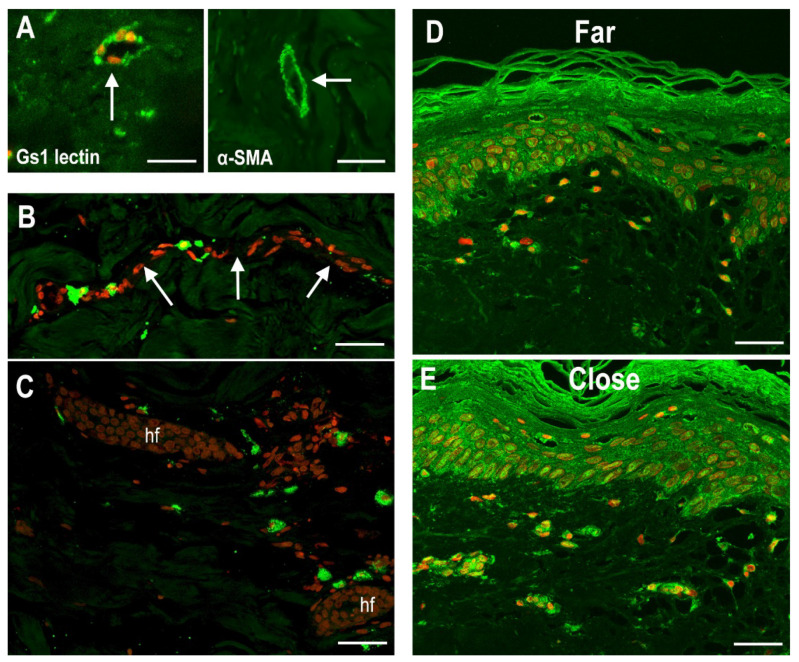
(**A**) Small blood vessels in the upper dermis (arrows) labeled by FITC-conjugate UEA lectin and anti-α-SMA antiserum. Mast cells, labeled by FITC-conjugated avidin, are located along blood vessels (arrows) (**B**) and around hair follicles (hf) (**C**). Visually, those located in proximity to the wound appeared to contain fewer fluorescent granules, suggesting that cell activation and granule release had occurred. (**D**,**E**) Dermal fibroblasts expressing the activation marker HSP47 appear to be more numerous in tissue areas close to the wound. HSP47 is also expressed by epidermal keratinocytes. Nuclei are counterstained in red with propidium iodide; bars = 100 μm.

**Figure 9 ijms-23-14123-f009:**
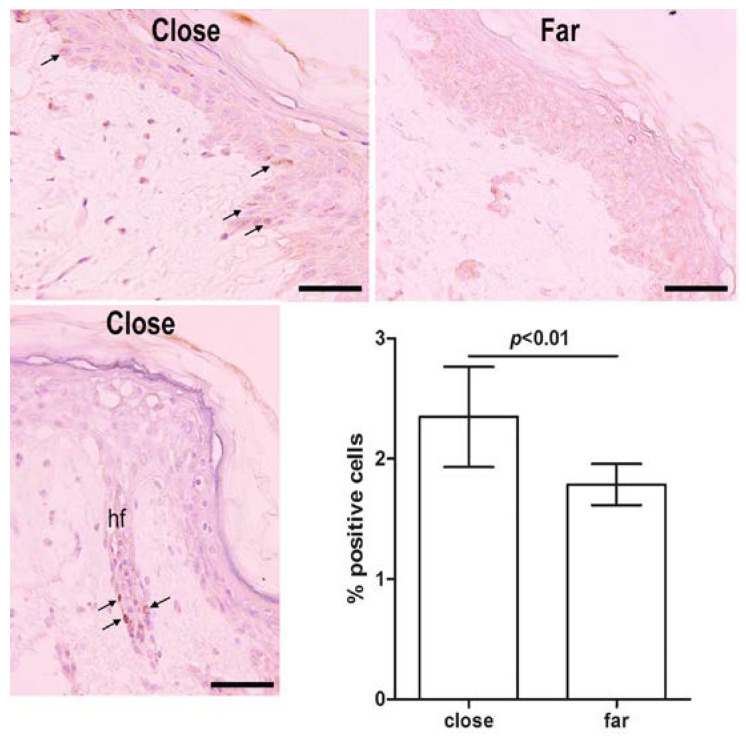
Immunostaining to reveal Ki67-positive proliferating cells in the skin specimens after long-term incubation in enriched medium, freezing and thawing. Positive cells (arrows) were slightly, albeit significantly, more numerous close to the wound than far (3–5 mm) from it. Some of them can be seen in the bulge of a hair follicle (hf). Ki67 immunoperoxidase staining; bars = 100 μm; values are mean ± s.e.m., *n* = 4.

**Figure 10 ijms-23-14123-f010:**
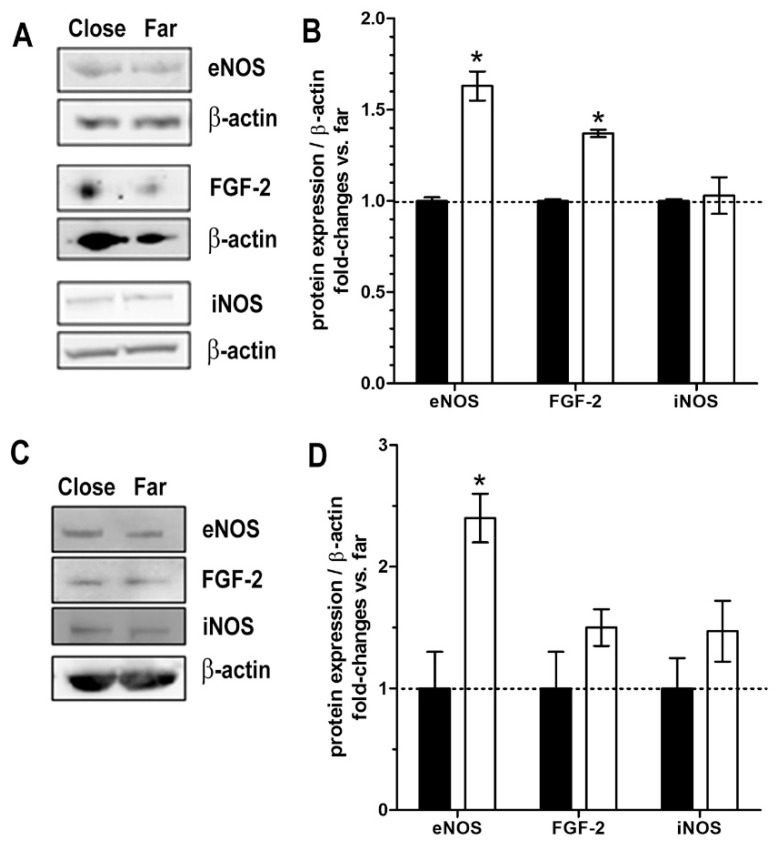
Western blots and densitometric quantitation for the endothelial integrity and functional markers FGF-2, eNOS and the inflammatory and tissue remodeling marker iNOS in the specimens of skin (**A**,**B**) and saphenous vein (**C**,**D**) taken in proximity to (close) or distant from (far, 3–5 mm) the surgical wound. β-actin was used as a loading invariant protein and assumed as reference control. (**A**,**C**) Representative blots of 3 independent experiments; (**B**,**D**) Bar graphs showing quantitation of the noted markers normalized to β-actin and expressed as fold-changes of the values measured in the samples taken far from the wound. Values are mean ± s.e.m., *n* = 4; * *p* < 0.05; no marks, not significant.

## Data Availability

Not applicable.

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
