# Peer review of "Optimization of an Ex-Vivo Human Skin/Vein Model for Long-Term Wound Healing Studies: Ground Preparatory Activities for the ‘Suture in Space’ Experiment Onboard the International Space Station"

_ijms, 2022, doi:10.3390/ijms232214123_

Round 1

Reviewer 1 Report

In this paper, Cialdai et al. developed an automated tissue culture chamber reproducing and a culture medium enriched with serelaxin and [Zn(PipNONO)Cl]. The experimental results confirm that ex-vivo human skin and vein specimens can be maintained viable for long time. This experimental work provides novel insights into construction of ex-vivo human skin model, and minor revisions are needed before they can be published on International Journal of Molecular Sciences. Suggestions are given below:

1. As described in the article, the human skin model under low gravity has not been studied yet. How is the dose or proportion of serelaxin and [Zn (PipNONO) Cl] determined?

2. Please explain the possible mechanism of microgravity damage to human skin?

3. Please elaborate on the microgravity environment and specific conditions?

4. The first abbreviation should be accompanied by the full name.

5. Under the same culture conditions, what are the differences between low gravity and normal gravity of human skin in vitro?

6. Please carefully check the format of the reference.

Author Response

We are grateful to this reviewer for favourable evaluation of our manuscript and useful suggestions to improve it. Accordingly, we have changed the text as summarized below:

1) The concentrations of serelaxin and Zn-nonoate have been chosen based on those exerting significant tissue protective effects in previous studies and in preliminary experiments conducted by us. This point has been added to the Methods section 4.3, lines 412-417.

2) The molecular mechanisms by which microgravity may influence human skin are only partly understood. This matter has been the object of some reviews, including one focused on the epidermis written by two of us [9]. Collectively, the available experimental data suggest that microgravity interferes at multiple levels with the regulatory signals which coordinate the different cell types involved in the repair process, thereby negatively affecting skin wound healing.

3 & 5) Indeed, the present experiment was designed to optimize the ex vivo tissue maintenance conditions in normal gravity in view of a forthcoming experiment to be performed on board the ISS, namely in microgravity. We focused our efforts on developing a culture technique that preserves the viability and function of tissues for over one month, which is the time needed to carry out the experiment on the ISS. Comparison of the present findings with those we hope to collect from the scheduled in-flight study could hopefully shed new light on the effects of microgravity on skin, wound healing in human tissues, and the involved mechanisms. In the current paper, we would prefer not to further discuss the possible effects of microgravity, since we could only offer speculations on it.

4) Thank you for warning: acronyms have been spelled out at first mention (namely: H&E, PSR, PAF, FITC, UEA, HSP47, FGF2, eNOS, iNOS).

6) References have been added with DOI, as per the journal’s style.

Reviewer 2 Report

Cialdai et al present a very interesting and new setup to study ex vivo skin grafts aboard the ISS.

My comments are:

- External air shall reach the tissue samples via a gas permeable membrane. Will this work under microgravity equally well as on ground, or are there differences? I wonder about air bubbles in the system. Have the authors maybe had the opportunity to analyze the technical aspects of the model under microgravity analogues such as a parabolic flight prior to establishing the model onboard the ISS?

- Did you check for tissue hypoxia? Or is there any evidence otherwise such as the expression of hypoxia induced proteins in the tissue?

- Do you plan to use tissue models without additives such as Serelaxin or would that lead to graft dysfunction? Assuming that microgravity will induce oxidative stress, is it possible that Serelaxin will prevent that from happening and thus might cover up some effects of microgravity?

Author Response

We are grateful to this reviewer for kind words of appreciation of our work and valuable suggestions to improve it. Accordingly, we have changed the text as summarized below:

1) The silicone membrane has been designed by Kayser Italia and manufactured by Taurus Sages (Rivoli, Italy). This type of membrane has already been used in past space experiment onboard the ISS (namely: BIOROCK, BIOASTEROID, BIOFILMS mission # 1 and 2). Therefore, the behavior of this membrane in microgravity conditions is known. The hardware used in our experiment has been developed through a long bottom-up engineering process, in which we and the developer company had to face and and solve the different issues raised at each step. In particular, the tubing and pumping system was optimized to prevent the formation of bubbles within the incubation chamber. Moreover, also the procedure to fill the hardware tanks with the culture medium have been adequately studied (the slower the filling takes place the lower the risk of bubble formation). This information has been added to the text (page 3, para. 2.1, lines 89-93). Nonetheless, the possibility of bubble formation cannot be completely excluded. We conducted a series of experiments to evaluate possible consequences of bubble formation on tissue biology and, from our results, such effects are negligible. Unfortunately, we could not test our model in real microgravity conditions.

2) In a previously published preliminary study [5] we did measure oxygen and its consumption in tissue cultures. In detail, as previously reported, oxygen consumption varied depending on tissue type, preservation conditions, time after collection, temperature and salinity. In general, oxygen consumption in tissue samples was ≤ 0.5 µmol/ml/h (salinity ranging from 0 to 10 g/l). The thickness (0.2 mm) and exchange surface (3350 mm2) of the membranes used has an oxygen permeability of 16.2 cm3/h, whjch largely fulfills the metabolic needs of the incubated tissues. This information has been added to the text (page 3, para. 2.1, lines 94-102).

3) Due to the complexity of the scheduled experiment at the ISS, we have taken all measures to preserve and extend tissue viability.  Our culture technique is based not only on biochemical factors (serelaxin, Zn-nonoate and the other substances contained in the culture medium), but also on biophysical factors, namely the modeling of physiological tensile strength in the tissues. At the beginning of our study we did perform experiments in which we compared: 1- samples maintained in culture medium without serelaxin and Zn-nonoate and without application of mechanical stress,   2- samples maintained in culture medium with serelaxin and Zn-nonoate but without application of mechanical stress, 3- samples maintained in culture medium without serelaxin and Zn-nonoate but with application of mechanical stress, 4- samples maintained in culture medium with serelaxin and Zn-nonoate and with application of mechanical stress. The best results (longer preservation of tissue viability and function) was obtained with the culture conditions applied in experiment #4. In particular, the unsatisfactory results obtained without serelaxin and Zn-nonoate are reported in figs. 2C, 3C, where histological findings show poor tissue preservation. The point on microgravity-induced oxidative stress is very thoughtful: indeed, serelaxin is known to have anti-oxidant properties (see ref. [7] Nistri et al. 2020) and could add this beneficial effect to further improve tissue preservation in microgravity conditions on board the ISS. This interesting point has been added to the discussion (page 13, lines 316-319)